# Examining the Composition of the Oral Microbiota as a Tool to Identify Responders to Dietary Changes

**DOI:** 10.3390/nu14245389

**Published:** 2022-12-19

**Authors:** Kirstin Vach, Ali Al-Ahmad, Annette Anderson, Johan Peter Woelber, Lamprini Karygianni, Annette Wittmer, Elmar Hellwig

**Affiliations:** 1Department of Operative Dentistry and Periodontology, Center for Dental Medicine, Faculty of Medicine and Medical Center, University of Freiburg, Hugstetter Straße 55, D-79106 Freiburg, Germany; 2Institute of Medical Biometry and Statistics, Faculty of Medicine and Medical Center, University of Freiburg, Stefan-Meier-Str. 26, D-79104 Freiburg, Germany; 3Clinic for Conservative and Preventive Dentistry, Center of Dental Medicine, University of Zurich, Plattenstrasse 11, CH-8032 Zurich, Switzerland; 4Department of Medical Microbiology and Hygiene, Institute of Medical Microbiology and Hygiene, Faculty of Medicine and Medical Center, University of Freiburg, Hermann-Herder-Straße 11, D-79104 Freiburg, Germany

**Keywords:** compositional data, responder, nutrition, microbiome, oral health

## Abstract

Background: The role of diet and nutrition in the prevention of oral diseases has recently gained increasing attention. Understanding the influence of diet on oral microbiota is essential for developing meaningful prevention approaches to oral diseases, and the identification of typical and atypical responders may contribute to this. Methods: We used data from an experimental clinical study in which 11 participants were exposed to different dietary regimens in five consecutive phases. To analyse the influence of additional nutritional components, we examined changes in bacterial concentrations measured by culture techniques compared to a run-in phase. A measure of correspondence between the mean and individual patterns of the bacterial composition is introduced. Results: The distance measures introduced showed clear differences between the subjects. In our data, two typical and three atypical responders appear to have been identified. Conclusions: The proposed method is suitable to identify typical and atypical responders, even in small datasets. We recommend routinely performing such analyses.

## 1. Introduction

In recent years, nutrition’s role in the prevention of oral diseases has gained increasing interest. To date, the oral cavity is one of the best-studied microbiomes, with a total of 392 taxa for which at least one reference genome is available, and a total number of almost 1500 genomes in the oral cavity [1]. Although some bacteria are known to be associated with oral diseases, such as *Fusobacterium* with periodontitis and *Neisseria*, *Streptococcus mutans*, *Lactobacillus* spp. with dental caries, and *Fusobacterium nucleatum* with oral cancer, it is difficult to determine in detail which oral bacteria are considered physiological in healthy conditions [2,3]. Sharma et al. and Segata et al. [4,5] show that the oral microflora of healthy persons mainly contain aerobes and obligate anaerobes of the genera *Streptococcus*, *Veillionella*, *Actinomyces*, *Neisseria*, *Candida* spp. and *Rothia mucilaginosa*.

Dietary factors have a large impact on the microbiota of the oral biofilm and can cause disruptions in the homeostasis of bacterial species in this complex network [6]. The consequences of frequent consumption of various foods, such as simple carbohydrates, dietary fibre, and certain vegetables, have been investigated in various studies [7,8,9,10,11,12,13,14,15]. The effects of nutrition on the oral microbiota have so far mainly been studied through in vitro experiments, tests of individual ingredients, animal experiments, or the analysis of epidemiological data. An in vivo study showed that the proportion of oral streptococci such as *Streptococcus gordonii*, *Streptococcus sanguinis*, and *Streptococcus parasanguinis* increased as a result of frequent consumption of rock candy, whereas some physiological members of the oral microbiome such as *Haemophilus* spp. and members of the phylum *Proteobacteria* generally decreased [7,16]. These results were in accordance with the ecological plaque hypothesis and with earlier reports that correlated *Haemophilus* spp. and the phylum *Proteobacteria* with the physiological microbiota and oral health [14,17,18,19,20,21].

When analysing the impact of dietary changes, it is common to present the mean values of individual changes for a series of bacteria [22,23], yielding a profile of mean values, whereby this type of presentation implicitly suggests that this profile is the typical profile for all individuals. In particular, it leads to the assumption that future participants will also display this average profile. While there may in fact be some participants whose profiles are similar to the average profile, others will have a different profile. From the perspective of the implicit assumption, the former can be considered typical responders in the sense that they respond to the dietary change as expected, while the latter are classed as atypical responders. In this paper, we present an approach that allows for quantifying the deviation of an individual profile from the average profile. The approach is applied to a longitudinal study in which several dietary changes are investigated so that the deviation can be examined several times. In this way, atypical responders can be identified as participants who repeatedly deviate from the expected profile. These participants need to be further examined and can provide additional insights.

We used data from a study in which the influence of changes in various nutritional components on the overall microbiota of the oral biofilm was investigated using splint systems on which the oral biofilm was cultured in situ. Culture techniques were then used to analyse the bacterial composition of the oral biofilm [7]. Both the individual variability in the changes of individual bacteria and the overall composition of the microbiome were studied previously [24,25]. In this study, the response behaviour of the participants is examined in more detail.

## 2. Materials and Methods

### 2.1. The Data

We examined a study funded by the German Research Foundation (Deutsche Forschungsgemeinschaft [DFG]) with 11 participants, in which the influence of additional standardised dietary components on the microbiota of the oral biofilm was investigated over 15 months. The study was conducted according to the guidelines of the Declaration of Helsinki and approved by the Ethics Committee of the University of Freiburg (No. 237/14).

The participants—with an average age of 32 years and good general and oral health—went through five phases, each lasting three months. In addition to the normal diet, uniform additional dietary components were administered in the form of sucrose (phase II), dairy products (phase III), and vegetables (phase IV). Figure 1 (motivated by an earlier version in [24]) visualizes the study design. Phase I was an introductory phase in which the participants’ dietary habits were not changed. In phase II, the participants ate an additional 10 g of rock candy daily, letting small pieces of 2 g melt on the tongue five times per day between meals. In phase III, the additional component consisted of dairy products (150 g of plain yoghurt three times a day and 100 mL of long-life milk twice a day, both with 1.5% fat). In phase IV, 500 g of vegetable puree per day was given to the participants, and finally, in phase V, the participants continued to eat according to their individual regular diets.

An in situ splint system with individual acrylic maxillary splints was used to obtain samples of the dental plaque of the participants [26]. The splint system was only removed at regular mealtimes and when performing dental hygiene and then placed in 0.9% NaCl solution. It was worn during the consumption of the phase-specific supplementary food and participants were instructed to eat slowly to expose the food mush to the oral cavity for several minutes. During all phases, the oral biofilm was allowed to grow on embedded enamel plates for seven days. Furthermore, the extraction of the oral biofilm followed the same procedure and was repeated three times in all phases. The biofilm was first allowed to grow on the embedded enamel plates for seven days, after which the splint system was removed to analyse the biofilm and cleaned. After seven days, the splint system was then reattached for another seven days. Standardised toothbrushes and uniform toothpaste with a sodium fluoride content of 1450 ppm were used by the participants for oral hygiene.

For all phases, the biofilm obtained was evaluated using the culture technique with an average of three measurements within four weeks per phase. Based on the phenotypically assessed and counted bacterial colonies, the number of colony-forming units (CFU) per ml in the original sample was calculated. More detailed information on this procedure is provided in [16,24,25].

A major focus of this article was to investigate the influence of additional standardised dietary components on modifications of the oral microbiota. According to the results of [24], only disjunct bacterial groups for which the percentage of values above the detection limit was greater than 75% were considered. The detection limit was used as the value in cases where values were below the detection limit. Figure 2b (taken from [24]) and Appendix A provide information on the relevant bacterial groups and the colour scheme we used in some of our figures. In the following, we use the term bacterium as a synonym for a bacterial group.

### 2.2. Analytic Strategy

For these types of microbiome data, it is common to use their log-10-transformed form. To draw conclusions about the behaviour of one single bacterium, its fraction of the total concentration of all bacteria is analysed. For each bacterium b=1,…,B, participant i=1,…,I, and phase *p*, this fraction of the total concentration is denoted by $f_{ip}^{b}$.

Since we are interested in investigating the effects of the addition of various dietary components to the overall diet on the microbiome, we presented the changes in concentrations compared to the run-in phase. This means that for each bacterium *b* and phase *p* the mean change $\mu_{p}^{b}$ of its contribution to the total concentration is computed by averaging the individual changes $\Delta_{ip}^{b}=f_{ip}^{b}-f_{i1}^{b}$, hence $\mu_{p}^{b}=\frac{1}{I}\sum_{i=1}^{I}\Delta_{ip}^{b}$.Hence, a value greater than 0 for a bacterium means that its quantity has increased on average compared to all other bacteria. The bacterium that has increased the most compared to the rest thus receives the largest positive value. Bar charts representing the various bacterial species were used to graphically represent the distribution of changes.Secondly, we examined how well the individual patterns match the mean pattern. To quantify the correspondence of the individual pattern with the mean pattern, the average squared distance was computed. In detail, a measure of the distance between the mean pattern $\mu_{p}^{b}$ and the individual pattern $\Delta_{ip}^{b}$ for the bacteria-specific mean changes of bacterium *b* and participant *i* can then be computed by $\tau_{ip}=\sqrt{\frac{1}{B}\sum_{b}{\left(\Delta_{ip}^{b}-\mu_{p}^{b}\right)}^{2}}$ for the bacteria b=1,…,B to get a measure for each phase *p*. To avoid overoptimistic results, $\mu_{p}^{b}$ was replaced by $\mu_{(-i)p}^{b}$, i.e., when computing the population average for participant *i* the value for participant *i* was omitted.Thirdly, based on the measure τ defined in this way, we defined participants as typical responders if their individual patterns match well with the mean pattern in all phases. Consequently, atypical responders are defined as those participants who repeatedly deviate from the expected profile. Specifically, participants are classified as typical (atypical) responders according to the distribution of τ if their value is smaller (larger) than the 40% (60%) percentile of τ in all phases.

The statistics program STATA (StataCorp LT, College Station, TX, USA, Version 17.0) was applied for all analyses. Bar charts, line plots, and heat maps were used for the graphical presentation of the results.

## 3. Results

In Figure 2, we present the mean values of the average changes in comparison to phase I for phases II–V [24]. In order to detect phase-specific patterns in the change of the bacterial spectrum, the bacteria within each phase are sorted according to the mean values of the average changes. However, it can be observed that the abundance of some bacteria tends to increase (*Campylobacter* spp., *Rothia* spp.) or decrease (*Neisseria* spp., *Capnocytophaga* spp.) across all phases, while other bacteria differ in the direction of change across phases (e.g., HACEK, *Fusobacterium* spp., *Gemella Granulicatella Abiotrophia species pluralis*).

Figure 2a describes a mean pattern of bacteria across all individuals. For the interpretation of the population variation in the composition, one can ask how well the individual patterns match this mean pattern. In Figure 3 the individual patterns are shown together with the mean pattern. A coloured line represents the values of each participant (1 red, 2 yellow, 3 cyan, 4 brown, 5 green, 6 purple, 7 lime, 8 orange, 9 light blue, 10 blue, and 11 lavender). The bacteria-specific mean value was drawn in with a black, slightly thicker line. For each phase, the bacteria are arranged according to their mean value in descending order. The numbers on the x-axis thus do not indicate the number of bacteria, but the range. For certain participants, at least some bacteria show larger deviations from the mean, e.g., in participant 7 (lime) in phase III and participant 6 (purple) in phase V. Overall, a large scattering of the individual changes can be observed, i.e., the mean patterns should not be regarded as valid for almost all participants.

However, it is also possible to quantify the correspondence between the individual patterns and the mean pattern. The defined measure τ describes the average squared distance between the mean pattern and the individual pattern for the bacteria-specific mean changes. Thus, a participant whose pattern is close to the mean pattern will have a τ that is close to zero. To examine this in more detail, we illustrate this for participants 1 and 6 in phase V in Figure 4. Participant 1 shows a very small distance to the (black) centre line for all bacteria, resulting in a small value of τ, while participant 6 always has a larger distance and thus the τ value is approximately three times as high.

When calculating these measures for each phase and subject (Table 1), it can be seen that the degree of agreement varies between subjects and phases. Some participants appear to follow the trend (1, 2) while others deviate from it (6, 8, 10, 11). This corresponds to the visual impression in (Figure 3).

When analysing the correspondence between the mean pattern and the individual patterns, the question arises whether there are participants who tend to follow the mean trend and others who tend to deviate across all phases. In other words, one might ask whether it is possible to identify typical and atypical responders. Figure 5 graphically shows the correspondence between the individual pattern and the mean pattern. The 20%, 40%, 60%, and 80% percentiles were used as cutpoints, i.e., the white boxes indicate a τ around the median. Strong red fields mark large gaps, i.e., where the individual values do not match the mean values at all, while the strong green fields indicate a good match. Looking at the values across all phases, two typical responders (1, 2) and three atypical responders (6, 8, 11) can be identified.

Having identified potential atypical responders in this way, it is advisable to again examine their individual profiles to gain a better understanding of the nature of the non-response. For example, atypical responders may always respond more markedly to dietary changes than the average, they may always respond more visibly to certain bacteria, or the pattern of deviance may change from phase to phase. In addition, the common deviation across phases may be related to a particular pattern at the beginning of the study, e.g., lower values for all bacteria considered. Figure 6 shows the individual profiles of bacterial concentrations of atypical responders in each phase (a) and the profile of mean changes compared to the baseline (b). In terms of the deviation of the individual profiles from the mean profiles (panel (b)), none of the atypical responders shows a common pattern between phase II and phase V. In particular, there is no tendency for the gains to be closer to 0, indicating non-compliance with the diet. Furthermore, the profiles at baseline (top of the panel (a)) show no systematic tendencies towards lower or higher values. However, the profiles of the raw measurements in the subsequent phases indicate that participant 8 frequently displays measurements below the detection limit, which may have contributed to their deviation from the average profile.

## 4. Discussion

In the present study, the change in microbiome composition due to various additional dietary changes was investigated using the culture technique. While *Campylobacter* spp. and *Rothia* spp. showed a tendency to increase during the phases, *Neisseria* spp. and *Capnocytophaga* spp. decreased. It was found that HACEK, *Fusobacterium* spp., *Gemella*, *Granulicatella*, *Abiotrophia species pluralis* and *Streptococcus* spp. responded differently to the additional dietary components in each phase. The fact that HACEK are fastidious microorganisms could explain why they react faster than other bacteria to changing environmental conditions [27]. The large increase in *Campylobacter* spp. could be explained by the fact that formiate can be utilised by *Campylobacter* spp., which *Actinomyces* spp. and *Streptococcus* spp. produce during glycolysis [28,29,30]. Not surprisingly, a different response to the dietary change was observed in the individual participants [24]. The methods presented for comparing individual patterns with mean patterns allowed us to get a picture of responders who, on average, responded as expected to the dietary change. The heat maps provide a good graphical representation of this.

It is difficult to verify whether the participants differ in terms of their dietary requirements on the one hand, and their wearing of the splint on the other hand. In terms of content, the question arises as to how the presence of atypical responders can be explained. In this context, it cannot be ruled out that there were problems with compliance, i.e., dietary requirements were not strictly adhered to, or the splints were not worn consistently. However, an inspection of the participants’ diaries indicates that they were very compliant in terms of adhering to the nutrition provided in the various phases. Looking at the individual profiles of atypical responders, as shown in Figure 6, participants who do not undergo any changes should have a profile close to the 0 line for changes whereas those who respond individually should show a similar pattern over time. For the atypical responders, there appears to be an individual variation that varies from phase to phase. Due to the large diversity of the salivary proteome, saliva may also have played a role in the atypical responders [31] and thus biological phenomena are also conceivable. However, it is evident that other factors not considered here also play a role, including demographic factors such as gender and age, but also psychological factors such as the motivation of the participants to change their diet [32,33]. One limitation of the study is the small number of cases, as, through the inclusion of more participants, subgroup analyses could have provided information about the causes of typical or atypical responses. The method presented could be used in future studies to learn more about the individual characteristics of typical and atypical responders to provide targeted individual dietary advice. Further studies, including a combination of different microbiological analysis methods, are needed to understand the long-term effect of diet on oral biofilm. However, such studies should also consider that there may be marked variations in the participants’ responses to a given diet.

## 5. Conclusions

It is possible to get an indication of typical and atypical responders with respect to dietary changes, even in small datasets.

## Figures and Tables

**Figure 1 nutrients-14-05389-f001:**
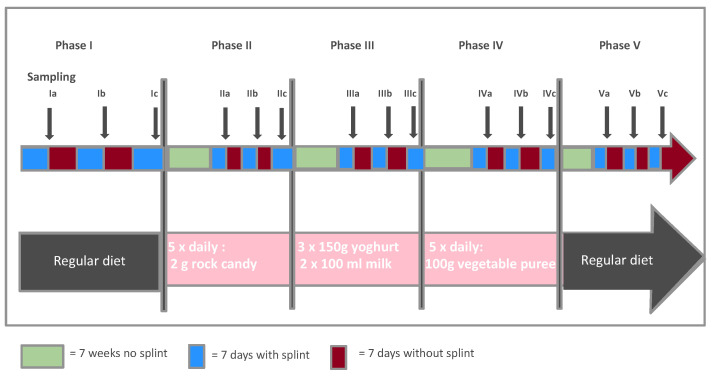
Description of the study design.

**Figure 2 nutrients-14-05389-f002:**
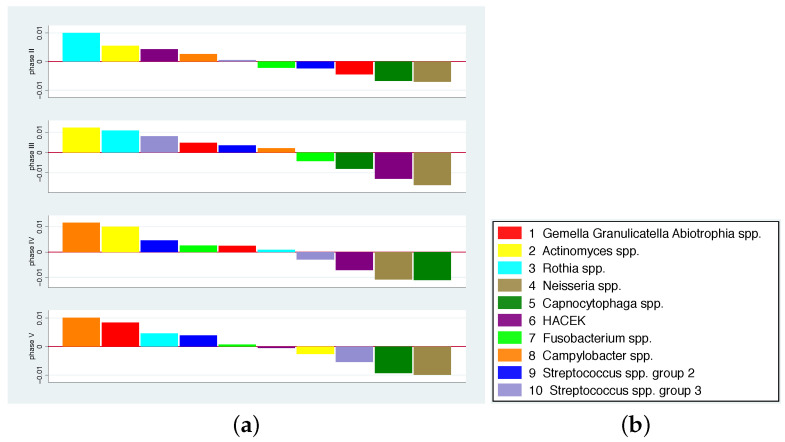
(**a**) Sorted mean values of the average changes for bacteria 1 to 10 for phase II (top) to phase V (bottom). (**b**) Colour scheme for the bacterial groups; spp. = species pluralis.

**Figure 3 nutrients-14-05389-f003:**
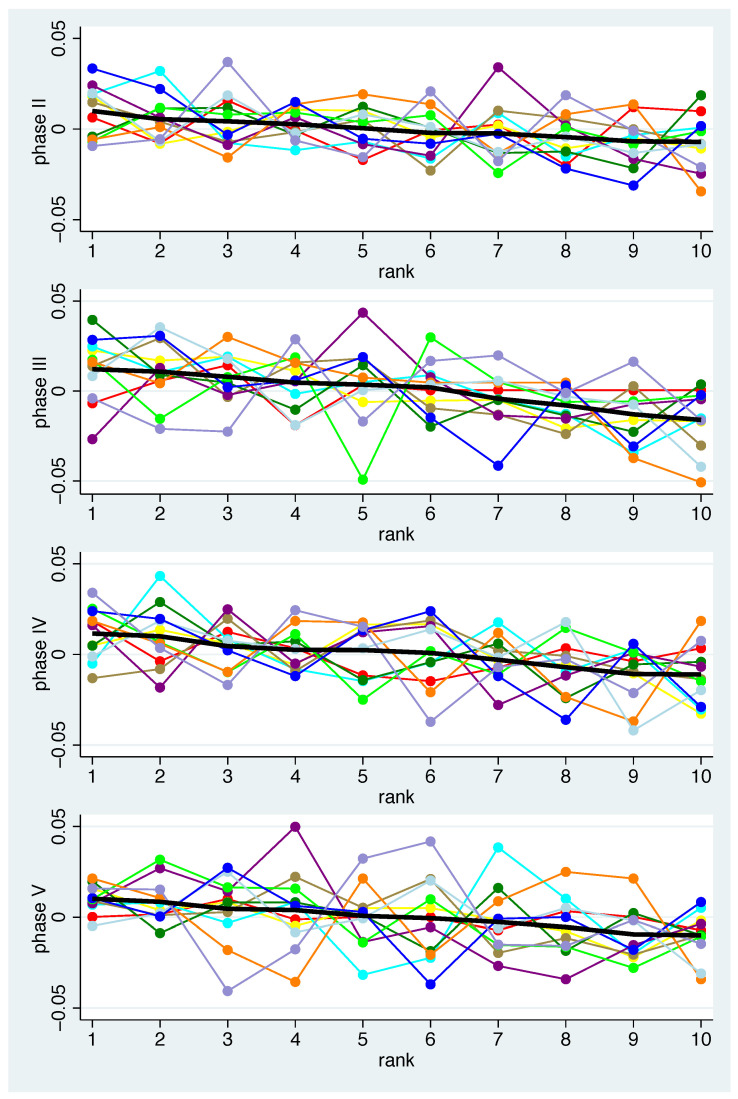
Individual mean changes for all bacteria and phases. Black line: population mean. The bacteria in each phase are in descending order according to their mean value.

**Figure 4 nutrients-14-05389-f004:**
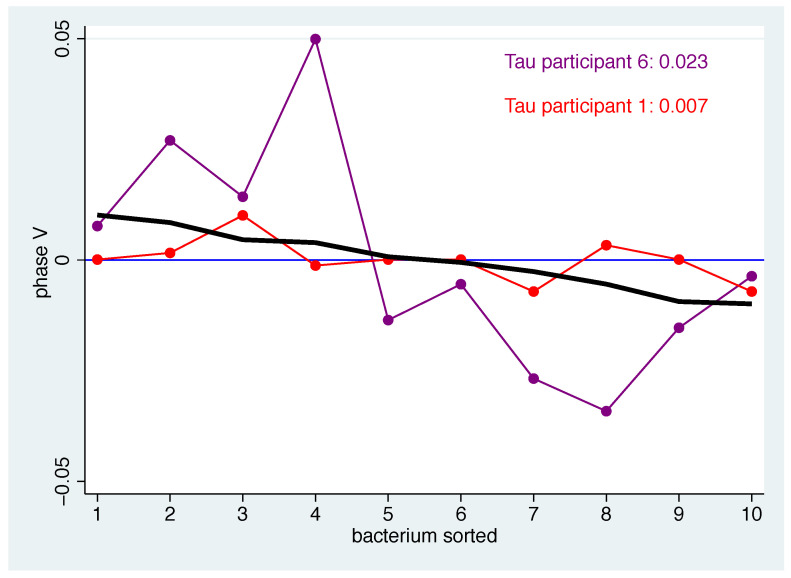
Individual mean changes for participants 1 (red) and 6 (purple) for all bacteria in phase V. Black line: population mean. The bacteria in each phase are in descending order according to their mean value.

**Figure 5 nutrients-14-05389-f005:**
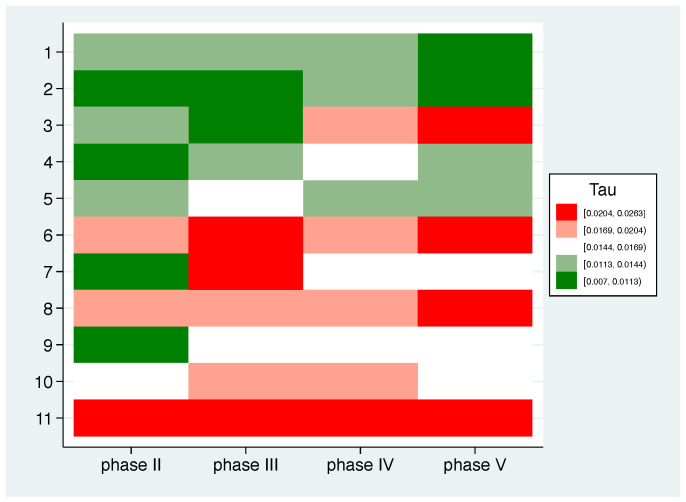
τ for all 11 participants acrosss phases II–V.

**Figure 6 nutrients-14-05389-f006:**
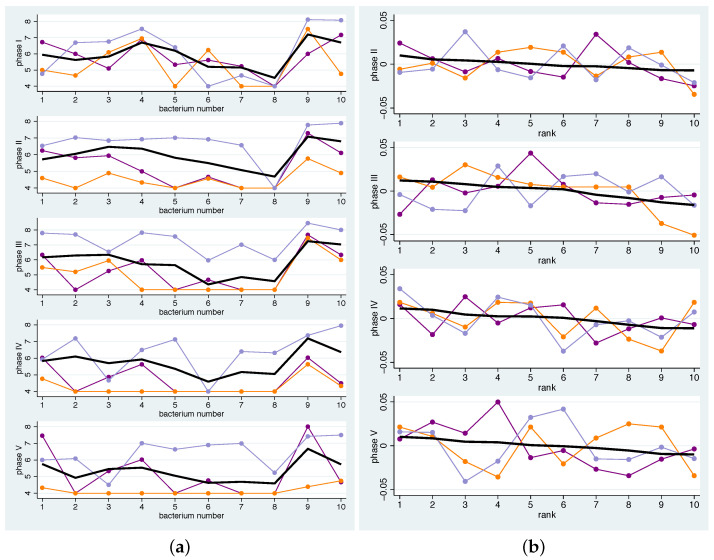
Individual patterns for atypical responders 6 (purple), 8 (orange), and 11 (lavender). Black line: population mean. (**a**) Individual bacterial concentrations of all bacteria in phases I–V. For the bacterial colour scheme used, see Figure 2. (**b**) Individual mean changes for phases II–V. The bacteria in each phase are presented in descending order according to their mean values.

**Table 1 nutrients-14-05389-t001:** Estimates of τ for participants **P1**–**P11** and mean values of all participants for phases **p II**–**V**.

	P1	P2	P3	P4	P5	P6	P7	P8	P9	P10	P11	Mean (SD)
**p II**	0.014	0.008	0.014	0.011	0.014	0.017	0.011	0.019	0.009	0.016	0.020	0.014 ( 0.004)
**p III**	0.014	0.009	0.010	0.014	0.016	0.021	0.024	0.018	0.016	0.020	0.024	0.017 (0.005)
**p IV**	0.012	0.011	0.019	0.015	0.012	0.017	0.015	0.020	0.016	0.018	0.021	0.016 ( 0.003)
**p V**	0.007	0.007	0.022	0.013	0.013	0.023	0.015	0.026	0.015	0.017	0.026	0.017 ( 0.006)
**all**	0.012	0.009	0.016	0.013	0.014	0.019	0.016	0.021	0.014	0.018	0.023	0.016 ( 0.005)

## Data Availability

The data are available on reasonable request from the authors.

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
