# Peer review of "Examining the Composition of the Oral Microbiota as a Tool to Identify Responders to Dietary Changes"

_nutrients, 2022, doi:10.3390/nu14245389_

Round 1
Reviewer 1 Report
Vach et al. present an interesting work that describes the oral microbiome composition as a tool to identify diverse dietary practices in the population which potentially carries translational as well as industrial implications.
This reviewer has only one recommendation:
Please include a graphical/schematic representation of the experimental setup (i.e. different phases described in the methods section). I believe this will be helpful for the readers.
Author Response
We thank the reviewer very much for the thorough reading of the manuscript.
An additional figure was added to explain the experimental setup.
Reviewer 2 Report
COMPOSITION OF THE ORAL MICROBIOTA AS A TOOL TO IDENTIFY RESPONDERS TO DIETARY CHANGES
Manuscript ID: nutrients-1973843
Nutrients
The aim of this in vivo longitudinal study was to analyze the influence of additional nutritional components during daily diets on the oral microbiota. The topic discussed is innovative and the results are very interesting, but the study design was not presented in detail. The authors did not provide enough methodological information to render the study reproducible. Unfortunately, the manuscript in its current form is not suitable for publication, but given the innovative topic and very interesting results, correction as suggested is recommended.
Abstract
- Please rewrite the results and conclusions more clearly
Introduction
- The following sentence should be provided more clearly: " This results in a profile of mean values, and this type of presentation implicitly suggests that this profile is the typical profile of all individuals. In fact, there may be some participants whose profile is similar to the average profile and others who have a different profile.”
- Several statements are lacking in references to the literature.
- It is recommended to argue which oral bacteria are considered physiological in healthy conditions.
Material & Methods
- How were the study subjects selected? Were they comparable in age and oral condition?
- The section should be enhanced by including detailed information about the method of sampling and bacterial culture.
Discussion
- References to the literature are scarce and should be added to the section.
- The authors should provide hypotheses to explain what was observed in the results. What could be the factors that influenced the subjects' atypical response?
Figures and Tables
- Check the refuse "Actiomyces spp." in Figure 1
- Insert a color legend in Figure 2, removing the reference to Fig.1.
In its current form the paper is unpublishable. However, authors are invited to submit it again once form improvements are introduced.
Author Response
Please find the point-by-point answers in the attached file.

Reviewer 3 Report
This generally interesting and innovative manuscript should be language-edited, some phrases are to be rewritten to sound more scientific and convincing. This also concerns the final sentence "It is possible to get a hint of typical and atypical responders with respect to dietary changes even in small datasets". Some recent relevant publications on the medical and dietary implications of the microbiota could be added to the reference list, e.g., Oleskin, A. V. and Shenderov, B. A. (2020). MICROBIAL COMMUNICATION AND MICROBIOTA-HOST INTERACTIONS: BIOMEDICAL, BIOTECHNOLOGICAL, AND BIOPOLITICAL IMPLICATIONS. Hauppauge (New York): Nova Science Publ.
Author Response
We thank the reviewer very much for the thorough reading of the manuscript.
According to the suggestions, some recent relevant publications have been added to the introduction and discussion.
Round 2
Reviewer 2 Report
I thank the authors for the changes made, only other minor adjustments are recommended:
- Health-associated bacterial species include Streptococcus salivarius, and Rothia mucilaginosa (Segata N, Haake SK, Mannon P, et al. Composition of the adult digestive tract bacterial microbiome based on seven mouth surfaces, tonsils, throat and stool samples. Genome Biol. 2012;13(6):R42. doi:10.1186/gb-2012-13-6-r42). Please add.
- Importantly, Fusobacterium nucleatum is associated with periodontitis and oral cancer (Pignatelli P, Romei FM, Bondi D, Giuliani M, Piattelli A, Curia MC. Microbiota and Oral Cancer as A Complex and Dynamic Microenvironment: A Narrative Review from Etiology to Prognosis. International Journal of Molecular Sciences. 2022;23(15):8323. doi:10.3390/ijms23158323).
- If available, it would be interesting to include a photo of the splint system used to obtain dental plaque samples.